# A Novel TMR Cantilever-Based Bi-Directional Flow Sensor for Agricultural and Domestic Applications

**DOI:** 10.3390/s25185915

**Published:** 2025-09-22

**Authors:** Anwar Ulla Khan, Ateyah Alzahrani

**Affiliations:** 1Department of Electrical Engineering Technology, College of Applied Industrial Technology, Jazan University, Jazan 45142, Saudi Arabia; 2Department of Mechanical and Industrial Engineering, College of Engineering and Computing in Al-Qunfudhah, Umm Al-Qura University, Makkah 24381, Saudi Arabia

**Keywords:** cantilever, ESP32 board, flow measurement, noninvasive measurement, TMR sensor

## Abstract

This article introduces a novel, cost-effective, noninvasive sensing mechanism for measuring water flow rate. It employs two tunneling magnetoresistance (TMR) sensors (analog and bi-polar), a magnet, and a stainless-steel cantilever. The TMR sensors are installed outside the insulating water pipe. A magnet is fixed at the free end of the cantilever and integrated into the pipe system. The cantilever’s deflection corresponds to the flow rate, with an analog TMR sensor measuring the bending angle. This bending angle, in either direction of the cantilever’s deflection, is captured through the analog voltage from the TMR sensor. The output from the analog TMR sensor is an analog voltage that directly reflects the strength of the magnetic field. An ESP32 microcontroller records the voltage from the analog TMR sensor, converts it to flow rates, and utilizes the bi-polar TMR sensor to ascertain the flow direction. A prototype sensor was developed and tested in a laboratory-scale setup to validate the effectiveness of the sensing mechanism. This prototype demonstrated a worst-case accuracy of 1.0% across flow rates of 0 to 1.5 m^3^/h for both the forward and reverse flow directions. The response and recovery times of the sensor are approximately 470 ms and 592 ms for forward and 487 ms and 625 ms for reverse direction flow. Also, hysteresis errors of 1.84% and 2.06% have been calculated for both flow directions. Notably, the sensing element does not contain any rotating components or require electrical connections to the cantilever for measurement. These attributes potentially lead to lower maintenance requirements and a longer lifespan for the sensor.

## 1. Introduction

Measuring water flow is crucial for maximizing both water and energy efficiency in a water supply system. Water flow meters play a vital role in tracking water flow rates in residential and agricultural water supply systems [1]. Selecting a flowmeter is contingent on the particular application and should take into account factors like required cost, accuracy, operating conditions, and consumption trends [2]. Common types of flow meters used in industrial settings include Coriolis, turbine, magnetic, and ultrasonic models [3].

Flow meters can be categorized into intrusive (or invasive) and non-intrusive (or noninvasive). The primary sensing element in intrusive flow meters makes contact with the water [3]. The design and dimensions of this sensing element affect the pressure drop that occurs with the flow. On the other hand, non-intrusive sensors do not create any pressure drop, typically boast a longer lifespan, and require minimal maintenance. Popular types of non-intrusive flow meters include electromagnetic and ultrasonic models. However, many existing non-intrusive flow meters tend to be costly for residential and irrigation use and have limited effectiveness in measuring low flow rates [4]. There is a growing need for affordable flow sensors that offer adequate accuracy and feature a design without rotating parts, which contributes to their durability and low maintenance requirements, especially in the residential and irrigation markets.

When discussing invasive meters, one example is a low-cost water flowmeter utilizing orifice plates designed for residential use, as mentioned in [5]. Although orifice plates are still prevalent in industrial settings, they result in a considerable pressure drop. The turbine flowmeter is another widely used low-cost water flow sensor [6,7]. The speed of the turbine’s rotor can be detected through various sensing techniques, such as the Hall effect or wireless self-powered methods [8,9]. However, the long-term viability of this sensor is limited due to its moving components and inability to manage water with debris.

Cantilever-based flow sensors are predominantly used in microfluidics and are not suitable for measuring flow in large-diameter pipes for domestic and agricultural purposes [10,11,12]. In these sensors, the cantilever is positioned perpendicular to the flow and bends in response to the flow’s dynamics. The bending angle can be detected using either strain gauges or image-based techniques. However, strain gauges require a wired connection to the pipe [13], while image-based methods necessitate a transparent aperture and a clear line of sight. Additionally, their efficacy can be hindered by challenging environmental conditions. Naveen et al. [14] explain how to implement a cantilever-based flow sensor for large-diameter pipes using image processing for data analysis. Harija et al. [15] have mainly employed the image-based method to evaluate characteristics in a controlled lab setting. While this approach is repeatable and relatively simple to model, the requirement for a transparent pipe window makes the camera-based method less practical for broader implementation. Thus, a noninvasive transduction mechanism is crucial for accurately sensing the cantilever’s bending angle.

Capacitive and inductive proximity sensors are commonly utilized for detecting positions [16,17,18]. Capacitive sensors are affected by factors such as moisture, humidity, and temperature. In contrast, inductive or eddy current sensors are known for their reliability; their outputs remain unaffected by the presence of lubricants, moisture, dust, and similar conditions. Eddy current sensors operate by using a sensing coil that is energized by an alternating current (AC) signal to gauge the proximity of a target object. The inductance of this sensing coil changes depending on the distance to the target. It is noted that many of the sensors mentioned are primarily designed for applications involving low flow rates in large-diameter pipes and tend to be costly for use in residential and agricultural settings.

This article introduces an innovative flow sensor that utilizes tunneling magnetoresistance (TMR). It measures variations in electric resistance caused by changes in the external magnetic field around the pipe to determine the bending angle of a cantilever resulting from water flow. The proposed sensor’s structure, design, and assessment are discussed in Section 2, Section 3, Section 4 and Section 5.

## 2. Sensing Approach with TMR Sensor

Figure 1 presents a simplified diagram of the cantilever-based sensing element, which is equipped with a magnet and installed inside a pipe (insulating material) of diameter D. The cantilever, made of stainless steel, has a magnet attached to its free end. The fluid flow (q) is directed perpendicularly to the cantilever. The cantilever bends in conjunction with the attached magnet based on the flow rate [14,15]. The bend’s direction is influenced by the direction of the flow. Due to the bending of the cantilever (along with the magnet) caused by the fluid flow, the magnetic field around the sensing element will change, which is detected using the TMR magnetic sensors positioned outside the pipe. TMR sensors work in such a way that when the magnetic field around the TMR sensor changes, the tunneling magnetoresistors of the TMR sensor change, due to which a linear voltage output is found that is proportional to the magnetic field strength.

Two TMR sensors have been used to measure flow rate and the direction of flow: analog and bi-polar TMR sensors. The analog TMR sensor produces a linear voltage output corresponding to a magnetic field strength range of −10 to 10 Gauss, with a sensitivity of −20 mV/V/G and an average supply current of 1.5 mA. This sensor is used to measure flow rates. The bi-polar TMR sensor with a 10 Gauss operate/−10 Gauss release capability, a sensing frequency of 500 Hz, and a push-pull output, which consumes an average current of 1.7 µA, is utilized to determine the flow direction [19,20].

Figure 2 illustrates a cantilever beam anchored at point P with its endpoint Q free. The beam bends under a uniform distributed load (w), resulting in the greatest deflection angle at the free end B. The beam’s overall deflection (x) is estimated as a mean angle along its length (h).

A mathematical relationship between the deflection angle and the fluid flow rate can be derived from two key assumptions. First, considering the no-slip boundary condition, the fluid velocity near the pipe wall is very low. Second, in the case of turbulent flows within pipes that have an internal diameter of less than 100 mm, the fluid velocity is regarded as nearly uniform throughout the cross-sectional area of the pipeline [21].

The velocity distribution exhibits significantly more uniformity in turbulent flow than laminar flow. This characteristic allows for the assumption that the force exerted upon the sensing element is uniformly distributed across its surface. Such a uniform distribution is critical for ensuring the accuracy and reliability of sensor performance and for the comprehensive analysis of flow dynamics.

In the analysis of finite loads, the differential equation that describes the deflection curve can be approximated as a function of the bending moment. This second assumption holds true under the condition that the deflection values remain significantly less than the length of the beam and the flexural rigidity is considered constant. This study presents a theoretical approximation of the cantilever deflection mechanism, which is referred to as the simplified cantilever.

## 3. Mathematical Model

As noted earlier, when a uniformly distributed load is applied, the axis of the cantilever beam bends into a curve. The differential equation describing this deflection curve is(1)$$\frac{d^{2}x}{dz^{2}}=\frac{M}{EI}$$
where x represents the deflection at point z, while M and EI denote the bending moment of the beam and its constant flexural rigidity, respectively. The deflection in the sensing element of length h (m) caused by the uniformly distributed load w (N/m) can be calculated as follows:(2)$$x=\frac{wh^{4}}{8EI}=\frac{fh^{3}}{8EI}$$

The normal force ‘f’ on the cantilever, with an area A_p_ (m^2^) oriented perpendicular to the fluid flow, is expressed as(3)$$2f=D_{C}A_{p}\rho v^{2}$$

Here, ρ (kg/m^3^) and v (m/s) denote the fluid’s density and average velocity, respectively. D_C_, the drag coefficient (dimensionless), measures an object’s resistance in a fluid. The Reynolds number (Re) illustrates the flow patterns, distinguishing between laminar and turbulent flow [14]. A Reynolds number exceeding 4000 signifies turbulent flow. In this research, the Reynolds number varies from 4000 to 20,322, based on the pipe diameter and flow velocity range. For turbulent conditions, D_c_ is estimated to be a constant value of 2 [21]. The deflection angle is defined as the inverse tangent of deflection divided by the length of the cantilever, as shown in (4).(4)$$\varphi =\tan^{-1}\left(\frac{x}{h}\right)$$

The flow rate (Q) is related to the pipe’s cross-sectional area (A_D_) and the average flow velocity, as outlined in Equation (5).(5)$$Q=vA_{D}$$

The flow rate can be articulated based on two specific assumptions [14,15]. First, because of the no-slip boundary condition, the fluid flow near the wall of the pipe moves very slowly. For turbulent flows in pipes that are less than 26 mm in diameter, we generally assume that the flow speed is nearly the same across the entire section of the pipe. The way the fluid flows in turbulent conditions is much flatter compared to how it flows in calm, or laminar, conditions. This means that the force acting on the sensor in the pipe is treated as uniform.

Second, when there are finite loads, we can describe the curve of the deflection (bending) using a formula based on the bending moment. This assumption works well as long as the amount of bending is less than the length of the beam and the beam has a consistent ability to bend (called flexural rigidity).

Based on the assumptions, the flow rate can be written as(6)$$Q\propto \sqrt{\tan\varphi }$$

The flow rate is proportional to the square root of the tangent of the mean deflection angle, as given by (6). This relationship applies to turbulent flow conditions (Re > 4000, Q > ~0.3 m^3^/h). Below this flow rate, the transition to laminar flow invalidates the assumption of a uniform velocity profile, and the force on the cantilever follows Stokes’ drag law, where the force is directly proportional to the velocity, leading to a different correlation between the flow rate and deflection.

The magnetic field component in the direction of the TMR sensitivity axis can be written as (refer to Figure 3).(7)$$B_{x}\propto \left|B\right|\sin\varphi$$

Also, the output voltage (V) of the TRM sensor (RR111-1DC2-331) is linearly proportional to the applied magnetic field and can be written as(8)$$V\propto B_{x}$$

From expression (6) to (8), the relationship between output voltage (V) and flow rate (Q) can be written as(9)$$V\propto \left|B\right|\frac{Q^{2}}{\sqrt{1+Q^{4}}}$$

From (9), it can be seen that the relationship between the output voltage (V) of the TRM sensor and the flow rate (Q) is non-linear.

Moreover, Table 1 shows that the assumptions regarding turbulent flow and uniform force distribution hold true for Reynolds numbers greater than 4000, which relates to flow rates greater than approximately 0.3 m^3^/h for the specified pipe diameter.

## 4. Orientation of TMR Sensor

For a magnetic sensor to effectively detect a magnet, two key conditions must be satisfied, regardless of their orientation [20].

-The magnetic field component that crosses the sensor’s sensing element must be aligned with its sensitivity axis.-This aligned component’s strength, known as Bx, needs to exceed a certain threshold for the sensor to respond; specifically, in a digital TMR sensor, Bx must surpass the magnetic operate threshold, often termed Bop.

With these requirements defined, Figure 4 shows the alignment of the TMR sensor for measuring bi-directional flow rates. The TMR sensor has a thickness of 2.90 mm, whereas the magnet measures 5 mm thick (refer to Figure 5a). If the TMR sensor is placed directly beneath the magnet, as illustrated in Figure 4b, it will produce a consistent output voltage with minor deflections of the cantilever (the magnet). However, suppose the TMR sensor is slightly offset from the magnet along the *x*-axis, as shown in Figure 5b. In that case, the output will vary in both directions of the cantilever (back and forth), enabling it to detect flow in both directions.

## 5. Experimental Setup and Results

A prototype of a TMR cantilever-based flow sensor mechanism was fabricated and evaluated using a laboratory-scale flow line setup. Figure 6a,b illustrate the connection representations of the experimental apparatus intended for flow sensing in both forward and reverse directions. Figure 7a,b depict the experimental setup and the associated measurement unit. The experimental apparatus incorporates the bypass valves, the control valve, the prototype flow sensor, and a calibrated flowmeter, facilitating comparative analysis of the performance metrics obtained from each device. A transparent acrylic section has been integrated into the PVC pipeline to enable visual monitoring of cantilever deflection during the initial testing phase. It is important to highlight that this transparency is not essential for the operational performance of the sensor, as the TMR elements are externally mounted and compatible with a variety of non-ferrous pipe materials commonly utilized in agricultural and domestic systems, such as PVC, HDPE, and copper. Two TMR sensors are used to record the deflection and direction of the strip in the fluid flow. Forward and reverse bypass valves have been used to bypass the fluid during forward and reverse flow rate measurements. A precise control valve has been used to control the fluid flow rate. The measurement unit is composed of a stainless-steel cantilever with dimensions of 25 mm in length, 18 mm in width, and a thickness of 0.15 mm. This cantilever is securely positioned within an acrylic glass pipe section with an internal diameter of 26 mm. The sensing element has been selected as stainless steel due to its exceptional corrosion resistance properties. This material provides substantial advantages, as it is applicable in both non-corrosive and aggressive fluid environments. Furthermore, the durability of stainless steel ensures consistent and reliable performance across a range of applications. At the same time, its inert characteristics preclude any possibility of water contamination, rendering it an optimal choice for deployment in aquatic settings. A magnet is securely mounted at the free end of the cantilever, as illustrated in Figure 8a. The specific type of magnet employed in this setup is an N35 round disc neodymium (NdFeB) magnet with a countersunk hole. Countersunk-hole pot magnets are constructed from premium rare earth materials, offering a temperature resistance of up to 80 °C (176 °F) (refer to Figure 8b). These magnets are enhanced by a sophisticated three-layer Ni + Cu + Ni plating process, which significantly improves their resistance to rust and corrosion and reduces brittleness [22]. This results in neodymium magnets that exhibit greater strength and durability. The specifications are as follows: the magnets have a diameter of 20 mm, a thickness of 5 mm, and a central hole of 5 mm in diameter, as illustrated in Figure 8c. The north and south magnetic poles are oriented on the flat surfaces.

The TMR mix-sens Click add-on board [23], which incorporates TMR digital push-pull and analog magnetic sensors, was utilized to measure the direction and deflection of the cantilever. This board features three types of magnetic field sensors: two digital sensors and one analog sensor. The digital sensors include the RR121-1A23-311, which exhibits an omnipolar response, and the RR121-3C63-311, which operates with a bi-polar response. The analog sensor, identified as RR111-1DC2-331, delivers an output voltage proportional to the magnetic field intensity. A bi-polar digital sensor (RR121-3C63-311) was used to determine the direction of flow. The magnetic flux response diagram of an analog sensor and the bi-polar digital sensor are shown in Figure 9a,b.

The outputs from all sensors are accessible through mikroBUS I/O or analog pins. An ESP32 board was used to collect the analog voltage from the RR111-1DC2-331 and the digital output from the RR121-3C63-311 sensors, thereby facilitating accurate measurement, calibration, data logging, and analysis of the cantilever’s deflection.

The test involved varying flow rates from 0 to 1.5 m^3^/h. The sensor exhibited immediate responsiveness to changes in fluid flow, with data collection occurring at two-minute intervals to ensure a steady-state condition was reached. Each recorded data point represents the running average result of over ten measurements, specifically on cantilever deflection attributed to the fluid dynamics.

A first-order RC low-pass filter (R = 47 kΩ, C = 100 pF, fc ≈ 34 kHz) was used primarily for RF noise suppression on the analog line prior to the ESP32’s internal ADC, which was sampled at 500 Hz. Software averaging was applied to mitigate low-frequency ADC noise. This configuration effectively stabilized the DC measurement for steady-state calibration. Capturing high-speed transients would require a different filter and ADC architecture, as discussed in the limitations.

The output from the calibrated flow meter, which was considered the true flow rate, was recorded simultaneously with the help of the ESP32 board.

### 5.1. Sensing of Flow

Figure 10a,b illustrate the outcomes of testing the prototype sensor. This research involved assessing the sensor’s performance in both forward and reverse flow directions within the pipeline. The analog TMR sensor produces an analog voltage that corresponds to the cantilever’s deflection in each flow direction. As shown in Figure 10, the voltage rises when the cantilever starts to deflect, particularly at initial flow rates, due to variations in the magnetic field. The voltage variation across the entire scale is non-linear. The non-linear relationship between voltage and flow, as indicated in Figure 10, is a fundamental characteristic of the sensing mechanism, as anticipated by the mathematical model detailed in Section 3. As expressed in Equation (9), the correlation between voltage (V) and flow rate (Q) is inherently non-linear.

Within the response curve, two regions of reduced sensitivity are identifiable. First, at low flow rates (around 0–0.5 m^3^/h), sensitivity is limited due to a combination of the cantilever’s rigidity and the shift in flow regime. Below a flow rate of roughly 0.3 m^3^/h (Re < 4000), the flow changes to a transitional state before becoming laminar. In a laminar flow, the drag force is directly related to velocity (f ∝ v) rather than being proportional to the square of the velocity (f ∝ v^2^), and the velocity distribution is parabolic instead of uniform. This results in a force on the cantilever that is lower than expected for a specified flow rate, making it more challenging to achieve significant deflection. Second, at higher flow rates (approximately 1–1.5 m^3^/h), sensitivity diminishes as the cantilever nears its maximum deflection. It is crucial to differentiate between two phenomena: (i) The ‘soft’ saturation described in Equation (9), where the fluid dynamic force component that contributes to further bending reduces as the cantilever aligns with the flow. (ii) A ‘hard’ structural saturation point, which is not specifically modeled, at which the cantilever’s deflection gets close to the material’s yield strength, beyond which plastic deformation may happen. The operational range of 0–1.5 m^3^/h is intended to stay well within the elastic deformation limits of the cantilever.

Despite this inherent non-linearity, the sensor remains highly effective across the entire specified range. The subsequent subsection outlines the development of calibration models that successfully linearize this relationship, ensuring precise flow rate measurements.

The theoretical behavior detailed in Equation (9) can be effectively understood by examining its asymptotic properties. In scenarios where Q is significantly less than 1 m^3^/h, the term *Q*^4^ becomes inconsequential, resulting in a simplified relationship where voltage V is proportional to the square of the flow rate *Q*. This indicates a quadratic increase in voltage corresponding to the flow rate. Conversely, in cases where *Q* exceeds 1 m^3^/h, the relationship adjusts to reveal that voltage V is proportional to |B|, thereby demonstrating saturation where the output voltage stabilizes and becomes unresponsive to additional increments in flow rate. This anticipated behavior, characterized by an initial quadratic rise followed by saturation, closely aligns with the general profile of the measured sensor output, as illustrated in Figure 10.

### 5.2. Calibration Model and Results

Utilizing the experimental data, which consists of 12 data points for forward flow and 15 for reverse flow [24], three calibration models were developed to correlate the fluid flow rate (derived from the reference flowmeter) to the analog voltage of the TMR sensor, as depicted in Figure 11a,b. The selection of the calibration dataset size was informed by the need to strike a balance between statistical rigor and the constraints of practical experimentation. The highly deterministic and repeatable characteristics of the cantilever’s response to flow rate yield a high signal-to-noise ratio. Such attributes facilitate a robust characterization of the system’s behavior with a minimized dataset, thereby mitigating the risk of overfitting while effectively capturing the non-linear dynamics of the response. Figure 12 presents a flowchart that delineates the selection process for these calibration models. The mathematical relationships are established through the least-squares method, which identifies the best-fit curve by minimizing the sum of squared errors for each data point. For further details regarding the calibration models applicable to both forward and reverse flow directions, please consult Table 2, Table 3, Table 4, Table 5, Table 6 and Table 7. To validate the proposed model, a residuals plots are shown in Figure 13. The sensor prototype computes flow rates via these calibrated equations embedded within the ESP32 microcontroller board. A comparison of the estimated flow rate with the actual flow measured by the reference flowmeter is illustrated in Figure 14, which additionally displays the percentage error for each data point in the final output. The prototype sensor attains an accuracy level of less than 1% of full scale for both forward and reverse flow readings.

The chosen dataset size is further validated by the exceptional precision of the models generated, as evidenced by the notably low standard errors of the coefficient estimates (refer to Table 2, Table 3, Table 4, Table 5, Table 6 and Table 7) and the resultant accuracy exceeding 1.0% FS across the entire operational range, as illustrated in Figure 14. The data points were judiciously distributed to ensure comprehensive representation of the entire flow range, with particular emphasis on the lower-sensitivity regions near both the minimum and maximum deflection.

To ensure that the selected data size is adequate, a statistical power analysis was conducted, and the results are summarized in Table 8. The power analysis indicates that the available dataset is sufficient for developing a precise calibration model, as the correlation between voltage and flow rate is exceptionally strong (R^2^ > 0.999). Additionally, a significant signal can be reliably detected even with a limited amount of data. Additionally, the low-flow calibration’s strength is further supported by a noise analysis, which identified a minimum detectable flow of 0.00144 m^3^/h (3σ, τ = 1200 s). This indicates a detection ability that is almost two orders of magnitude under the usual operational threshold of 0.1 m^3^/h.

Key parameters such as response time, recovery time, repeatability, and hysteresis are crucial for ensuring the sensor performs reliably and resiliently in real-world applications.

Figure 15a,b depict the transient response curves for a forward flow change of 1.038 m^3^/h and 1.254 m^3^/h in the reverse direction. The sensor exhibits response times of approximately 470 ms and recovery times of about 592 ms, while for reverse flow, the response and recovery times are roughly 487 ms and 625 ms, respectively.

Figure 16a,b show the sensor’s repeatability at consistent flow rate levels across multiple cycles of water flow in both forward and reverse directions. The sensor exhibits high repeatability, characterized by a sharp rise in voltage output and an abrupt increase in flow, followed by a leveling off at a saturation value. Upon refreshing, the voltage output reverts to its original value.

The hysteresis error is represented as ${∆V}_{max}/\delta_{FS}$, with ${∆V}_{max}$ signifying the highest discrepancy between output voltage values at the same flow rate during both the increasing and decreasing phases of the flow rate change, $\delta_{FS}$ refers to the full-scale output voltage. The test results for both output voltage and flow rate phases are illustrated in Figure 17a,b, encompassing both forward and reverse flow directions. The largest difference between the increasing and decreasing phase data is observed at a flow rate of 0.64 m^3^/h, at which point ${∆V}_{max}=30\;mV$ for the forward direction and 0.422 m^3^/h, at which point ${∆V}_{max}=40\;mV$ for the reverse direction, leading to calculated hysteresis errors of 1.84% and 2.06%, respectively.

The significant parameters of the proposed sensing mechanism are summarized in Table 9.

Table 10 compares the proposed sensor with a commercial flow meter [14,15,25,26,27]. The proposed sensor offers several significant advantages over the target flow meter.

Firstly, sensitivity stands out as a crucial differentiator; the proposed sensor demonstrates superior sensitivity compared to the target flow meter, which utilizes a flat disc or sphere attached to an extension rod as its sensing element. This configuration is based on two main assumptions: (i) the extension rod’s cross-sectional area is substantially smaller than that of the drag element, and (ii) the drag element itself is much less than the pipe’s internal diameter. Consequently, existing target flow meters exhibit minimal deflection, resulting in reduced sensitivity. In contrast, the proposed method features a thin stainless steel flat plate as the sensing element, enabling more significant deflection due to its capacity to bend freely to a considerable angle.

Secondly, the proposed sensor can measure bi-directional fluid flow, which offers an advantage over the target flow meter typically designed for unidirectional flow measurements. The proposed sensor accurately assesses flow rates from 0 to 1.5 m^3^/h in a consistent turbulent flow regime. Additionally, using a corrosion-resistant flat plate expands the prototype’s versatility in clean and aggressive fluid environments. On the other hand, the target flow meter primarily caters to high-viscosity fluids, slurry, gas, and steam in filled pipes at flow rates reaching up to 4 m^3^/h while functioning under low Reynolds number conditions [28,29]. To contextualize the performance of the proposed sensor for its target agricultural and domestic applications, its key characteristics are compared with those of commercial flow meters in Table 10. The “Requirements” row defines the primary criteria for these markets, emphasizing low cost, sufficient accuracy, and operational robustness [4,5].

An examination of the specifications presented in Table 10 indicates that the proposed sensor meets several critical criteria for both domestic and agricultural applications. Notably, its projected cost is significantly lower than the target benchmark, its accuracy is maintained within the requisite sub-5% range, and its design is devoid of rotating components, contributing to its durability. Furthermore, the sensor is equipped with bi-directional flow detection, a feature that is not commonly found in low-cost meters.

However, it is essential to note that the operational flow range of the current prototype (0–1.5 m^3^/h) falls short of the upper limit of the desired range (0–5 m^3^/h). Additionally, the semi-intrusive cantilever design may pose a potential risk of clogging due to debris, particularly when compared to completely non-invasive alternatives. Addressing these identified limitations—specifically the flow range and debris resistance—will be essential in guiding future enhancements. Strategies to consider include expanding the sensor design to accommodate higher flow rates and integrating a protective inlet filter to mitigate these concerns.

When compared to the proposed sensor with MEMS-based cantilever sensors [11], MEMS cantilevers are highly sophisticated devices that typically outperform in response time and miniaturization due to their semiconductor-grade fabrication processes. The key distinction lies in the cost–performance trade-off. While MEMS cantilevers offer exceptional performance, they involve significant fabrication costs and can be delicate in certain applications. Our sensor, on the other hand, emphasizes cost-effectiveness, robustness, and ease of fabrication, making it suitable for applications where extreme miniaturization and rapid response times are less critical. Table 11 shows a trade-off (cost vs. performance) between the proposed sensor and MEMS-based cantilever sensors.

## 6. Discussion and Future Work

The functionality of the proposed sensor is dependent on two essential external conditions. Firstly, it is imperative that the pipe material is non-ferromagnetic, such as PVC, HDPE, copper, or brass, to allow for effective penetration of the magnetic field. The presence of ferromagnetic materials, such as carbon steel, would inhibit the field, rendering the sensor inoperable. Secondly, the liquid in question should exhibit effective diamagnetic or paramagnetic properties with very low magnetic susceptibility, such as water, to prevent any distortion of the magnetic field. Although the sensor is not designed for use with ferromagnetic fluids, this characteristic aligns well with its intended applications in water management across both agricultural and domestic contexts.

The laboratory results regarding the proposed sensor demonstrate significant promise; however, its performance in practical applications may be impacted by various environmental and physical factors that were not examined during the initial testing phase. This section presents a discussion on potential limitations and sources of error, along with recommendations for mitigation strategies and avenues for future research. Future work will also involve thorough numerical simulations using COMSOL Multiphysics 5.6 to investigate design parameter spaces further.

### 6.1. Impact of Temperature Variations

Variations in fluid temperature can influence the precision of the sensor through two main processes:

Fluid Properties: The density (ρ) and viscosity of water fluctuate with changes in temperature. This variation influences the drag force (as described in Equation (3)) acting on the cantilever for a specified flow rate, potentially resulting in modifications to the calibration curves presented in Figure 9.

Material Properties: The Young’s modulus (E) of the stainless-steel cantilever is influenced by a temperature coefficient. A change in E can modify the flexural rigidity (EI) of the beam, which in turn affects its deflection (as described in Equation (2)) for a specified load, potentially introducing measurement inaccuracies.

Mitigation Strategy: The most effective approach involves integrating a temperature sensor, such as a thermistor, into the measurement unit. This would enable the ESP32 microcontroller to utilize a temperature-dependent calibration model, represented as Q = f(V, T), to compensate for these effects in real-time.

### 6.2. Susceptibility to Magnetic Interference

The operational principle of the TMR sensor makes it sensitive to external magnetic influences.

Magnetic Particulates: The accumulation of ferromagnetic particles within the fluid may adhere to the neodymium magnet, resulting in alterations to its magnetic field profile. This change could subsequently cause a systematic drift in the output voltage of the sensor.

External Magnetic Fields: Electromagnetic fields generated by nearby motors, transformers, or other machinery could disrupt the TMR sensors, leading to inaccurate readings.

Mitigation Strategy: A filtration system is highly recommended for applications involving contaminated fluids to ensure the protection of the magnet. Utilizing a mu-metal enclosure to shield the sensor assembly can significantly mitigate the influence of external magnetic fields. Furthermore, implementing software-based baseline calibration under established zero-flow conditions can effectively address any slow drifts that may occur.

### 6.3. Scaling and Design Limitations

The existing prototype has been fine-tuned for a particular flow range (0–1.5 m^3^/h) and a pipe diameter of 26 mm.

Scaling the Flow Range: To accommodate significantly higher flow rates or larger pipe diameters, a redesign of the cantilever will be necessary. To prevent over-deflection, it will be important to utilize a stiffer cantilever, which can be achieved by increasing its thickness, reducing its length, or selecting a material with a higher Young’s modulus. It is essential to note that this enhancement may result in decreased sensitivity, underscoring the importance of carefully designing and precisely calibrating the sensor for its intended operational range.

Inherent Limitations: The cantilever’s semi-intrusive design results in a slight reduction in pressure and poses a potential clogging hazard in the absence of a filter. Additionally, the sensor necessitates separate calibration for various configurations.

Overcoming Limitations: The sensor is designed with a modular approach. While the external TMR readout system remains consistent, the internal cantilever can be adapted for diverse applications, ensuring the retention of its inherent cost advantages

### 6.4. Validation of Assumptions

The theoretical framework is founded on the assumption of a nearly uniform velocity profile within turbulent flow conditions. While this assumption serves as a standard and legitimate simplification for the specified operating range (Q > ~0.3 m^3^/h), it is not applicable in the case of laminar flow (Q < ~0.3 m^3^/h). In the lower flow region, the velocity profile exhibits a parabolic shape, resulting in a force distribution on the cantilever that diverges from the model’s assumptions. This factor contributes to the observed reduction in sensitivity at very low flow rates, as depicted in Figure 10.

### 6.5. Durability Tests

This research sets the foundation for the preliminary static calibration and precision of the sensor. A subsequent important phase, which is already underway, involves an extensive long-term durability assessment under ongoing operation with harsh fluids (such as saline) to measure drift and resistance to corrosion, which is crucial for industrial certification.

### 6.6. Long-Term Stability of NdFeB Magnets

The long-term stability of sintered NdFeB magnets in aqueous environments is a critical consideration for ensuring sensor longevity. While these magnets are effectively safeguarded with a [e.g., Ni-Cu-Ni] coating in compliance with industry standards, conducting a dedicated accelerated life testing study is essential for quantitatively predicting magnetic flux decay and assessing its impact on calibration drift over multi-year operational periods. This investigation has become a central focus of our ongoing research efforts.

### 6.7. Reliability

The long-term fatigue life of the cantilever is a crucial factor in ensuring its reliability. In light of the reviewer’s suggestion, we recognize the importance of conducting a study to establish the cantilever’s S-N curve and to characterize its failure modes through the use of SEM. This endeavor will be a central focus of our forthcoming research efforts.

## 7. Conclusions

This article presents an innovative, cost-effective, and noninvasive sensing scheme for a bi-directional flow sensor based on cantilever bending principles. All experiments were conducted under controlled laboratory conditions, simulating turbulent flow within a 26-mm-diameter pipe, using typical tap water as the working fluid. The sensor developed is suitable for both residential and industrial applications. The system employs two tunneling magnetoresistance (TMR) sensors—one analog and one bi-polar—to simultaneously measure flow rate and direction. Anchored within the pipe and equipped with a magnet at its free end, a cantilever experiences deflection proportional to flow dynamics. The analog TMR sensor accurately detects this bending angle. Notably, the design eliminates the need for electrical connections to the cantilever, enhancing its integration and minimizing complexity. The TMR sensors are strategically positioned to enable concurrent assessment of flow rate and directional flow. The prototype is calibrated for a flow range of 0–1.5 m^3^/h, achieving an impressive accuracy of less than 1%. Future work will focus on evaluating the sensor’s performance with conductive fluids and implementing the proposed mitigation strategies, which include temperature compensation and robustness testing against magnetic interference. Overall, this proposed sensor architecture presents a streamlined approach to flow monitoring systems; however, it is imperative to carefully consider its operational environment to ensure sustained accuracy over time.

## Figures and Tables

**Figure 1 sensors-25-05915-f001:**
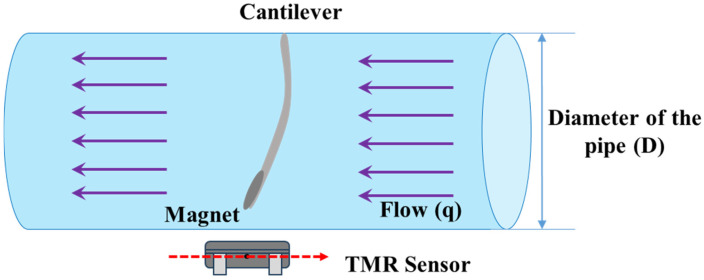
Conceptual diagram illustrating the cantilever-based sensing element integrated with a magnet and a TMR (tunneling magnetoresistance) sensor.

**Figure 2 sensors-25-05915-f002:**
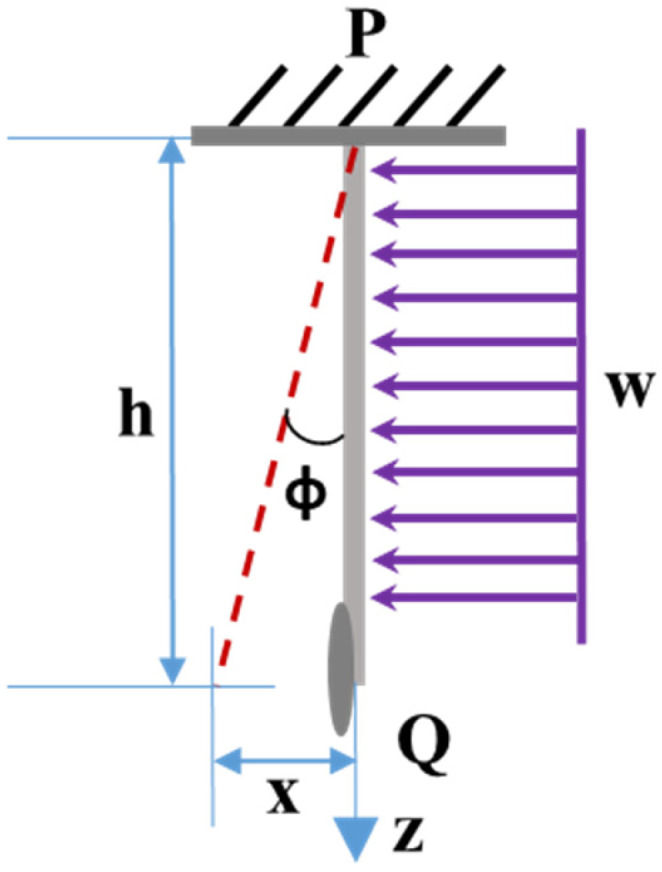
Deflection profile of a cantilever beam under uniform load.

**Figure 3 sensors-25-05915-f003:**
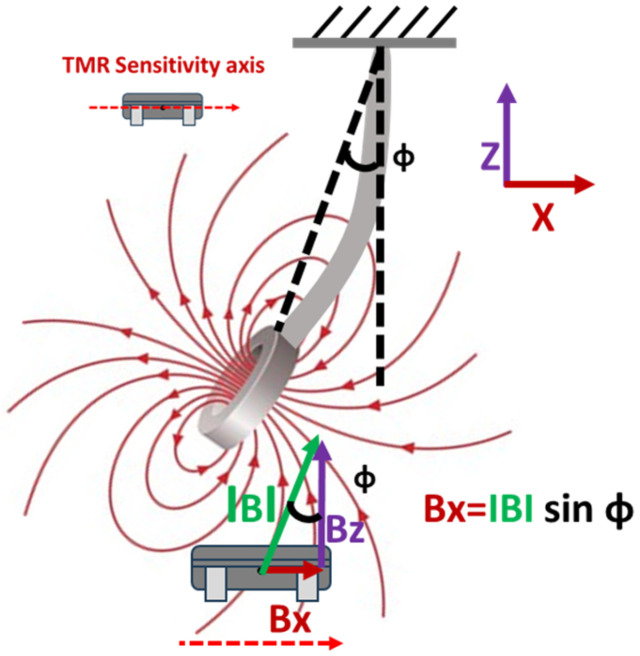
Component of magnetic field in the direction aligned with the TMR sensor’s sensitivity axis.

**Figure 4 sensors-25-05915-f004:**
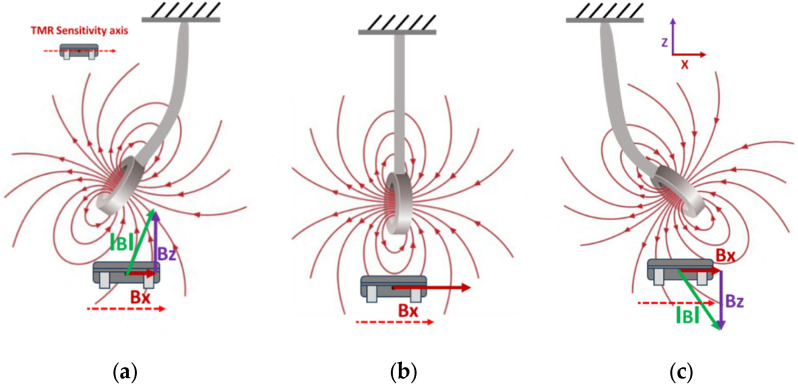
Orientation of the TMR sensor. (**a**) Change in the magnetic field of the magnet due to deflection in the forward direction aligned with the TMR sensor’s sensitivity axis. (**b**) Normal position of the TMR sensor. (**c**) Change in the magnetic field of the magnet due to deflection in the reverse direction aligned with the TMR sensor’s sensitivity axis.

**Figure 5 sensors-25-05915-f005:**
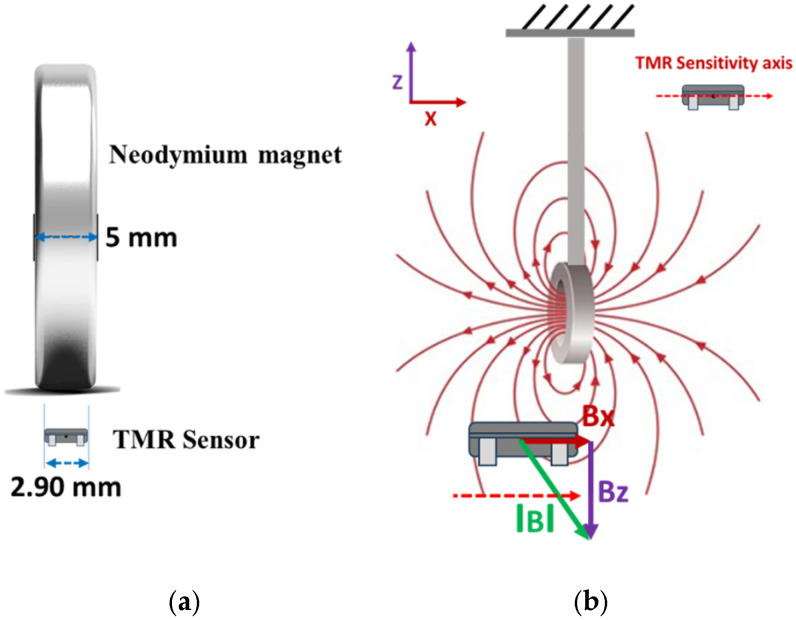
(**a**) Thickness comparison of neodymium magnet and TMR sensor. (**b**) Offset placement of TMR sensor along the *x*-axis.

**Figure 6 sensors-25-05915-f006:**
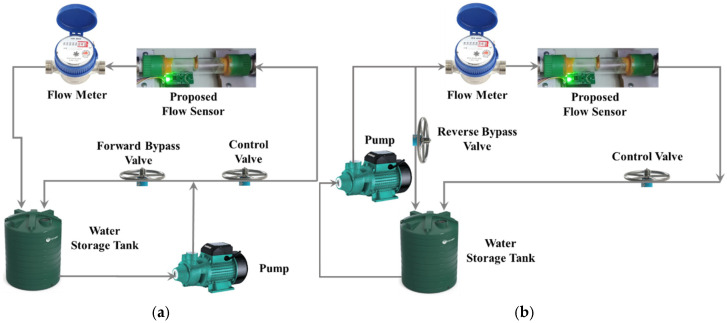
Connection diagram of the experimental setup, detailing two configurations: (**a**) sensing in the forward direction and (**b**) sensing in the reverse direction.

**Figure 7 sensors-25-05915-f007:**
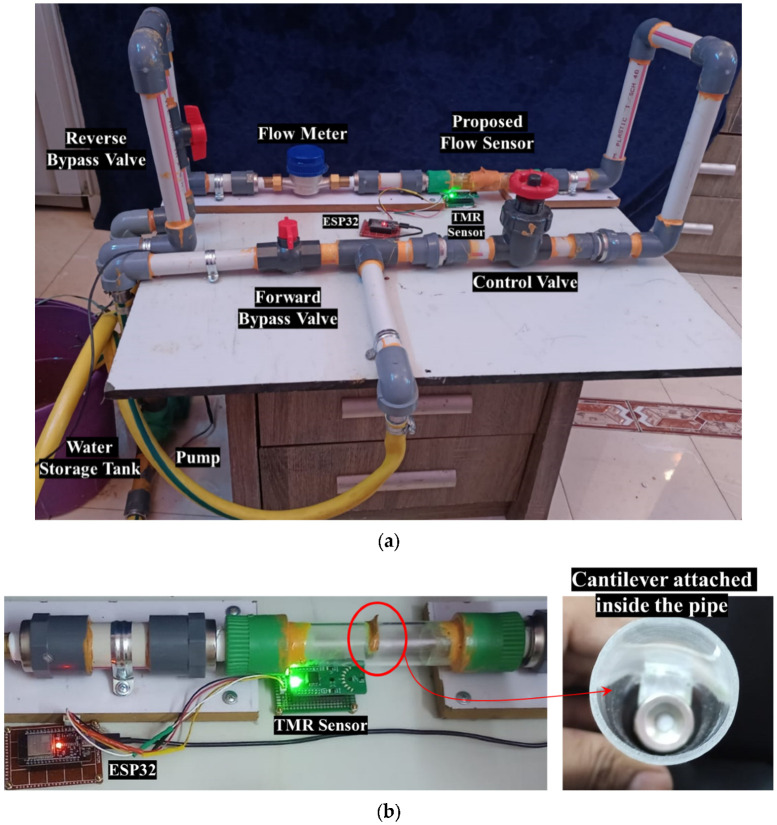
(**a**) An experimental setup to measure the flow in both directions, which can be configured per the connection diagram shown in Figure 5a,b. (**b**) Prototype flow sensor. The detailed view of the test section highlights the placement of the sensing element along with the noninvasive readout mechanism.

**Figure 8 sensors-25-05915-f008:**
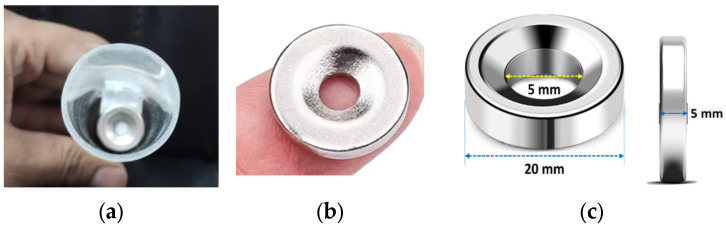
(**a**) Mounting of the magnet at the free end of the cantilever. (**b**) N35 round disc neodymium (NdFeB) magnet with a countersunk hole. (**c**) Dimensions of the magnet.

**Figure 9 sensors-25-05915-f009:**
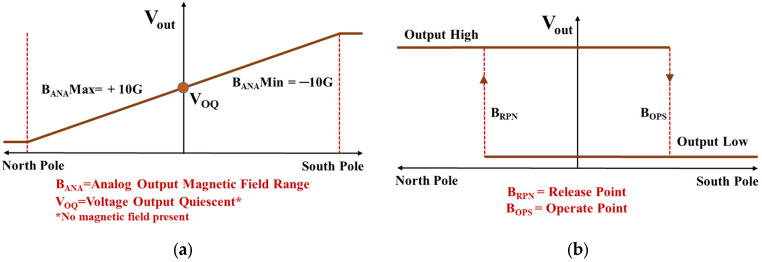
Magnetic flux response diagram. (**a**) Analog RR111-1DC2-331 sensor. (**b**) Bi-polar RR121-3C63-311 digital sensor (active low option).

**Figure 10 sensors-25-05915-f010:**
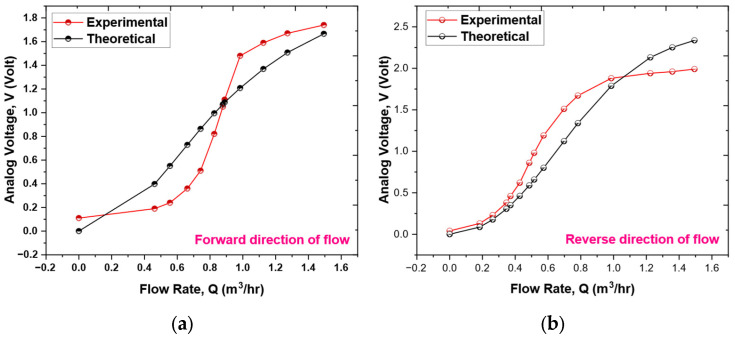
The plot depicts the sensor output voltage versus flow rate, featuring measured data, an empirical polynomial calibration curve, and a normalized theoretical curve from Equation (9). The theoretical curve shows the expected quadratic increase at low flow rates and a saturation at higher rates, confirming the model’s accuracy in representing the non-linear behavior of the sensing mechanism. (**a**) Forward direction of flow and (**b**) reverse direction of flow.

**Figure 11 sensors-25-05915-f011:**
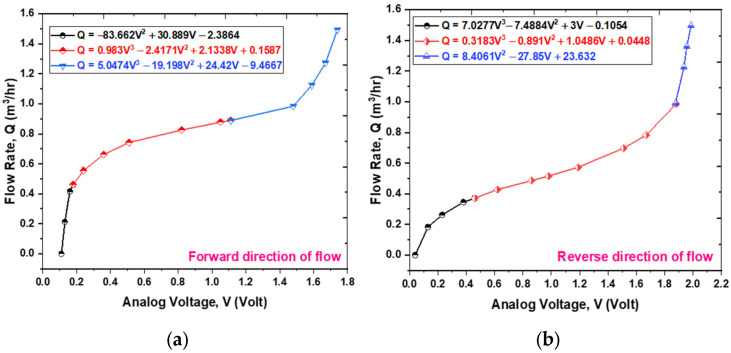
Calibration models with calibrated equations. (**a**) Forward direction flow values and (**b**) reverse direction flow values.

**Figure 12 sensors-25-05915-f012:**
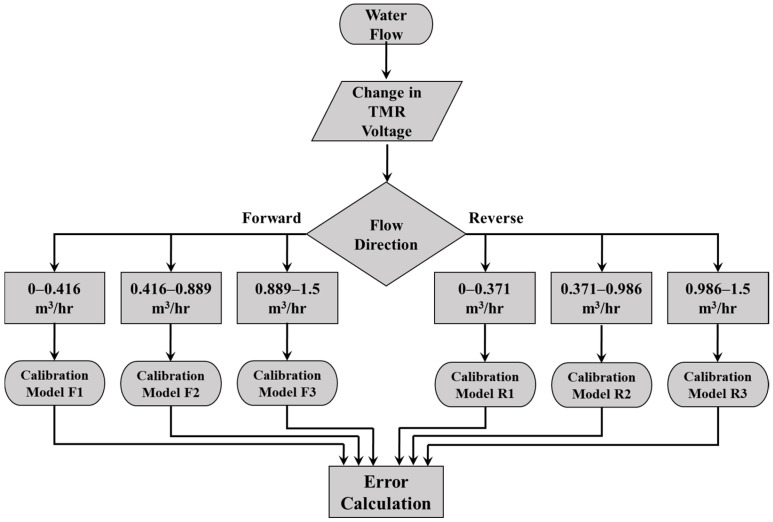
Flowchart detailing the criteria for selecting calibration models.

**Figure 13 sensors-25-05915-f013:**
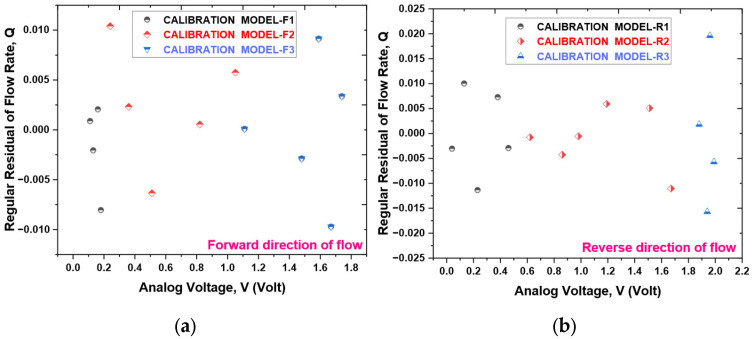
Regular residual of flow rate. (**a**) Forward direction of flow and (**b**) reverse direction of flow.

**Figure 14 sensors-25-05915-f014:**
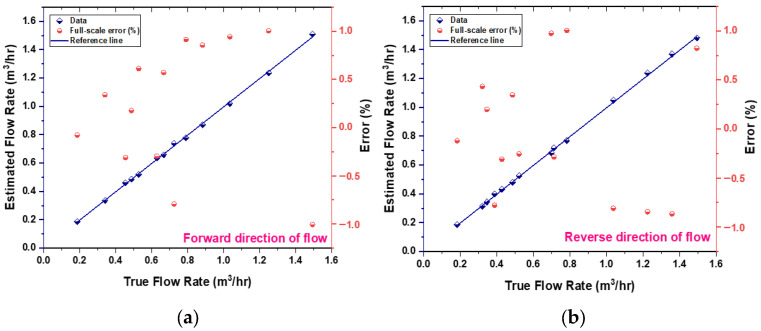
Estimated flow rate (from prototype flow sensor) compared to true flow rate (from reference flow sensor). (**a**) Forward direction flow. (**b**) Reverse direction flow. The indicated errors reflect the percentage of the full-scale error associated with the measurements.

**Figure 15 sensors-25-05915-f015:**
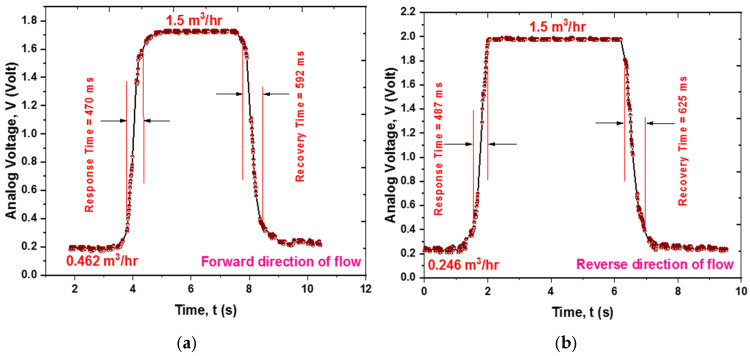
Transient response of the sensor mechanism: (**a**) for 1.038 m^3^/h flow change in the forward direction; (**b**) for 1.254 m^3^/h flow change in the reverse direction.

**Figure 16 sensors-25-05915-f016:**
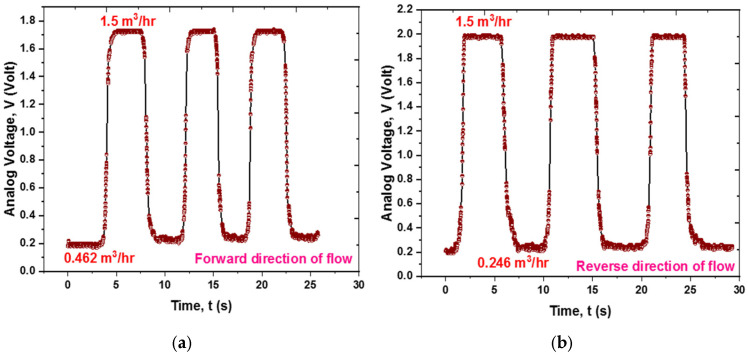
Repeatability of the sensor mechanism: (**a**) for 1.038 m^3^/h flow change in the forward direction; (**b**) for 1.254 m^3^/h flow change in the reverse direction.

**Figure 17 sensors-25-05915-f017:**
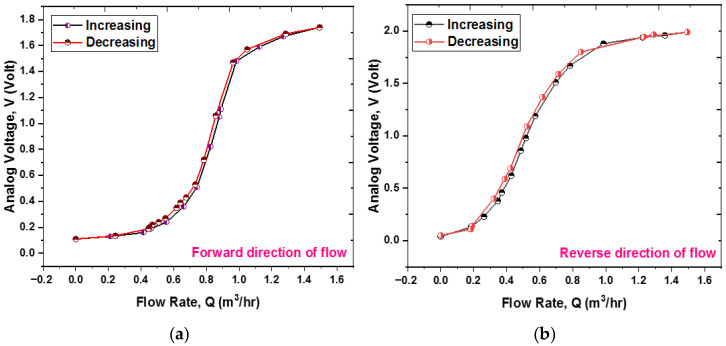
Hysteresis curve. (**a**) Forward direction flow. (**b**) Reverse direction flow.

**Table 1 sensors-25-05915-t001:** Flow rate vs. Reynolds number and resulting flow regime.

Flow Rate, Q (m^3^/h)	Reynolds Number (Re *)	Flow Regime
0	0	No flow
0.295	4000	Turbulent (Re > 4000)
0.36	5000	Turbulent
0.5	6774	Turbulent
1.0	13,548	Turbulent
1.5	20,322	Turbulent

* The standard equation to determine the Reynolds number (Re) for water flowing in a pipe (with a diameter of 26 mm) is $R_{e}\approx 13,548\times Q.$

**Table 2 sensors-25-05915-t002:** CALIBRATION MODEL-F1.

Coefficients	Value	Standard Error	R-Square
a_1f_	−83.662	3.158	1
b_1f_	30.889	0.918	
c_1f_	−2.3864	0.065	
Model-F1: Q = a_1f_V^2^ + b_1f_V + c_1f_			

**Table 3 sensors-25-05915-t003:** CALIBRATION MODEL-F2.

Coefficients	Value	Standard Error	R-Square
a_2f_	0.983	0.192	0.998
b_2f_	−2.4171	0.369	
c_2f_	2.1338	0.209	
d_2f_	0.1587	0.033	
Model-F2: Q = a_2f_V^3^ + b_2f_V^2^ + c_2f_V + d_2f_			

**Table 4 sensors-25-05915-t004:** CALIBRATION MODEL-F3.

Coefficients	Value	Standard Error	R-Square
a_3f_	5.047	1.896	0.999
b_3f_	−19.198	8.255	
c_3f_	24.420	11.808	
d_3f_	−9.4667	5.530	
Model-F3: Q = a_3f_V^3^ + b_3f_V^2^ + c_3f_V + d_3f_			

**Table 5 sensors-25-05915-t005:** CALIBRATION MODEL-R1.

Coefficients	Value	Standard Error	R-Square
a_1r_	7.0277	4.019	0.996
b_1r_	−7.4884	2.983	
c_1r_	3	0.619	
d_1r_	−0.1054	0.033	
Model-R1: Q = a_1r_V^3^ + b_1r_V^2^ + c_1r_V + d_1r_			

**Table 6 sensors-25-05915-t006:** CALIBRATION MODEL-R2.

Coefficients	Value	Standard Error	R-Square
a_2r_	0.3183	0.037	0.999
b_2r_	−0.891	0.128	
c_2r_	1.0486	0.137	
d_2r_	0.0448	0.044	
Model-R2: Q = a_2r_V^3^ + b_2r_V^2^ + c_2r_V + d_2r_			

**Table 7 sensors-25-05915-t007:** CALIBRATION MODEL-R3.

Coefficients	Value	Standard Error	R-Square
a_3r_	8.406	9.571	0.995
b_3r_	−27.85	36.990	
c_3r_	23.632	35.727	
Model-R3: Q = a_3r_V^2^ + b_3r_V + c_3r_			

**Table 8 sensors-25-05915-t008:** Summary table of power analysis (for forward direction flow).

Segment	Voltage Range	Model Order	Sample Size (N)	No. of Predictors (u)	R^2^	Effect Size (f^2^)	Power (1-β)
1	0.11–0.18 V	2nd	4	3	>0.999	~200,000	>99.99%
2	0.19–1.11 V	3rd	6	4	>0.999	~3500	>99.99%
3	1.12–1.74 V	3rd	4	4	1.000	~1,000,000	100%

**Table 9 sensors-25-05915-t009:** Prototype sensor system parameters.

Parameter	Forward Flow	Reverse Flow
Response time	470 ms	487 ms
Recovery time	592 ms	626 ms
Hysteresis error	1.84% (at 0.64 m^3^/h)	2.06% (at 0.422 m^3^/h)
Operating range	0–1.5 m^3^/h	0–1.5 m^3^/h

**Table 10 sensors-25-05915-t010:** Comparison between different water flow sensors.

Flow Sensor	Flow Rate (m^3^/h)	Functioning Method	Complexity	Rotating Elements	Accuracy [% Full Scale]	Cost [USD]	Applications
Requirements (Domestic/Agric.)	0–5	-	Low	No	<5	<100	Must handle water with some debris; low maintenance
Proposed work	0–1.5	Cantilever mechanism	Simple	No	0.5–1	20–30	The system detects bi-directional flow with a corrosion-resistant sensor.
Ultrasonic meter	2.5–35	Doppler’s shift	Complex	No	1–2	400–2740	Manage clean and corrosive fluids in fully occupied pipes.
Target flow meter	0–4	Measures the force on the target.	Complex	No	1–3	500–1000	This is for high-viscosity fluids with low Reynolds numbers in filled pipes.
Rotameter	1–14	Differential Pressure	Complex	Yes	1–2	55–400	Maintain clean fluids in filled pipes.
Turbine meter	4.5–35	Velocity	Complex	Yes	1–2	82–700	

**Table 11 sensors-25-05915-t011:** Trade-off (cost vs. performance) between the proposed sensor and MEMS-based cantilever sensors.

Aspect	MEMS Cantilever Sensors [11]	Our Sensor (This Work)	Implications & Suitable Applications
Fabrication Cost	Very High	Very Low	MEMS: High-volume production is essential to cover costs. Ours: Perfect for prototyping, custom designs, and low-volume use.
Unit Cost	Low	Extremely Low	Ours: Supports disposable, single-use, or widely distributed sensor networks.
Performance	Excellent	Good/Fit-for-Purpose	MEMS: Demands meticulous packaging and treatment. Ours: Appropriate for tough, unclean settings.
Robustness	Low	High	Ours: Quicker deployment times for experimental arrangements and integrated systems.
Design Flexibility	Low	Very High	MEMS: Incomparable for applications with limited space (such as implantables).
Ease of Integration	Complex	Simple	
Miniaturization	Excellent	Limited	Ours: Ideal for exploration, prototyping, and designing geometries tailored for specific applications without incurring retooling expenses.

## Data Availability

The original contributions presented in the study are included in the article, further inquiries can be directed to the corresponding authors.

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
