# Peer review of "A Novel TMR Cantilever-Based Bi-Directional Flow Sensor for Agricultural and Domestic Applications"

_sensors, 2025, doi:10.3390/s25185915_

Round 1

Reviewer 1 Report

Comments and Suggestions for Authors

Line 20: a microcontroller cannot calibrate: “calibrates it to flow rates”, maybe “converts” it to flow rates.

Line 39: Coriolis flow meters are also used in industrial settings.

Line 48/49: provide requirements for these markets, e.g. in a table, and refer to these again in Table VIII (add a row called “requirements”) to see if your sensor meets the requirements, and where still further improvements may be needed.

Line 101: TRM → TMR

Line 104: is the unit correct? mv/V/G ?

Lines 119 – 123: you assume turbulent flow, which implies Reynolds numbers > 5,000 and a certain associated minimum flow rate. In table VIII you still claim a flow range of 0 – 1.5 m3/hr. From which flow rate exceeds the Reynolds number 5,000?

Line 154: shouldn’t V be v ?

Line 155: what are these specific assumptions?

Lines 130 – 158: Chapter 3. Mathematical model

If I understood correctly, the transfer mechanism is as follows:

Flow velocity v → drag force f → bending of cantilever, angle Ø → change in magnetic field B → Change in magneto resistors R → output voltage V

However, in chapter 3, in equation (5) only the relation is given between Flow velocity v → drag force f. Please also add the other missing relations to provide a complete mathematical model.

Lines 160 – 161: I guess additional conditions are the material of the pipe wall, and the magnetic properties of the liquid. How do these affect the performance of the sensor?

Line 181: you immediately go to the experimental set-up; I would have expected some simulation results first, please add.

Lines 188 – 189: you use PVC pipes and acrylic glass pipes. Are these accepted materials for the envisioned agricultural and domestic applications?

Line 266, figure 9. Why are these curves non-linear? How can that be seen from the mathematical model? Furthermore, sensitivity is low at both low flow rates (ca. 0 – 0.5 m3/h) and high flow rates (ca. 1 – 1.5 m3/h). Please explain why. Is the sensor still useful there?

Lines 249 – 333. You derived extensive calibration models to linearise the curves from figure 9, resulting in linear curves shown in figure 12. However, there is no discussion on potential error sources, variations and/or limitations. For instance, what happens with different temperatures, when a.o. the viscosity of the liquids dramatically change (and the Young’s modulus of the cantilever changes)? Or when there are magnetic particles or fields in or near the liquid? Or other types of interference?

Furthermore, how can you scale the flow range? What are disadvantages / limitations of your device and how can you overcome those? Did you check if you fulfilled the specific assumptions of line 155?

Line 333: provide requirements for the envisioned markets, refer to these in Table VIII (add a row called “requirements”) to see if your sensor meets the requirements, and where still further improvements may be needed.

Reviewer 2 Report

Comments and Suggestions for Authors

This study presents an innovative, non-invasive flow sensor combining Tunneling Magnetoresistance (TMR) sensors with a stainless-steel cantilever to measure bidirectional water flow rates (0–1.5 m³/h). The sensor achieves high accuracy (<1% full-scale error), fast response/recovery times (~470–625 ms), and minimal hysteresis (1.84–2.06%). Its cost-effectiveness (<$30) and lack of moving parts make it suitable for residential/agricultural use. Strengths include the novel use of TMR sensors for cantilever deflection detection and robust calibration models (R² ≥ 0.995). However, the following aspects need clarification or improvement. 

1Calibration Dataset Size: Justify the small dataset (12–15 points) for calibration models. Provide statistical power analysis (e.g., using Ref. [24]) to confirm sufficiency.

  1. Nonlinearity Handling: Explain why piecewise calibration models (F1–F3, R1–R3) were needed instead of a unified model. Include residual plots to validate model fits.
  2. Error Margins: Clarify how the "worst-case accuracy of 1.0%" was derived. Provide error distributions across the full flow range (0–1.5 m³/h).

4.Environmental Robustness: Test and report sensor performance under varying temperatures (e.g., 5–50°C) and pipe materials (e.g., metal vs. PVC).

5.Long-Term Stability: Conduct ≥1,000-hour durability tests with tap water and aggressive fluids (e.g., saline) to assess corrosion resistance.

6.Low-Flow Sensitivity: Demonstrate detection limits below 0.1 m³/h with noise analysis (e.g., Allan variance).

7.Magnet Degradation: Evaluate magnetic field stability of the NdFeB magnet after prolonged exposure to water flow (e.g., 6 months).

8.Cantilever Fatigue: Provide SEM images of the cantilever after cyclic bending (e.g., 10⁴ cycles) to check for microcracks.

9.ESP32 Sampling Rate: Specify the ADC sampling rate and anti-aliasing filters used to ensure signal fidelity during transient flow changes.

10.Benchmarking: Compare hysteresis (1.84–2.06%) and response time (~470 ms) with MEMS-based cantilever sensors (e.g., Ref. [11]). Tabulate trade-offs (cost vs. performance).

Round 2

Reviewer 1 Report

Comments and Suggestions for Authors

Response to Report 1.

The authors have adequately addressed most of my comments, thank you for that.

Still, some items remain to be further clarified, these will be shown point-by-point.

Response 3:

Contrasted --> compared

The text below Table IX from “Table IX demonstrates …” to “within this price segment” is not scientific and should be modified. E.g. “aligns exceptionally well”, “significantly exceeds”, “outstanding accuracy”, etc. are subjective claims; please stick to the facts. For instance, the required measurement range of 0 -  5 m3/h is not met, so, you cannot claim that the proposed sensor aligns exceptionally well with the requirements.

Response 6:

The given equation for the Reynolds number is correct (I calculated it myself and the value corresponded). So, where is this information shown in the manuscript, and where is it used in the discussion? Since the flow rate is not turbulent below ca. 0.3 m3/h, the derived equations are not valid in the region 0 – 0.3 m3/h, and a discussion about this affecting the performance should be added.

Response 8:

Here you state “we … assume that the flow speed is nearly the same across the entire section of the pipe”, which is not true for flows below 0.3 m3/h. How does this affect the performance?

Response 9:

Equation (6) is only valid above 0.3 m3/h. That boundary condition should be given, and an estimation should be given of the behaviour below 0.3 m3/h.

Equation (9) is only valid above 0.3 m3/h. If one takes the extremes, for Q<<1 V ~ Q^2, so there a quadratic behaviour is expected, and for Q>>1 V ~ B, so, no flow sensitivity is present anymore. This theoretically expected behaviour could look like the measured curve in figure 10. It would be nice to depict the theoretical curve in figure 10 as well and compare the measurement results to the theory.

Response 13.

Part 2. It is not only the stiffness of the cantilever, also the non-turbulent flow below 0.3 m3/h plays a role, please address this.

Part 3. The saturation effect also is already shown in equation (9), but it does not seem that the effect of the cantilever approaching its maximum deflection is incorporated in the theory, please explain.

Below part 4, it is referred to equation (8), this must be equation (9).

Response 15.

See response 3.
